# *Phyllanthus emblica*: Phytochemistry, Antimicrobial Potential with Antibiotic Enhancement, and Toxicity Insights

**DOI:** 10.3390/microorganisms13030611

**Published:** 2025-03-06

**Authors:** Gagan Tiwana, Ian Edwin Cock, Matthew James Cheesman

**Affiliations:** 1School of Pharmacy and Medical Sciences, Gold Coast Campus, Griffith University, Gold Coast 4222, Australia; g.tiwana@griffith.edu.au; 2School of Environment and Science, Nathan Campus, Griffith University, Brisbane 4111, Australia; i.cock@griffith.edu.au; 3Centre for Planetary Health and Food Security, Nathan Campus, Griffith University, 170 Kessels Rd., Nathan 4111, Australia

**Keywords:** antimicrobial resistance, natural antibiotics, plant-based antimicrobials, combinatorial interactions, phytochemical profiling, secondary metabolites

## Abstract

*Phyllanthus emblica* Linn. (commonly known as Amla or Indian Gooseberry) is commonly used in Ayurvedic medicine to treat respiratory infections, skin disorders, and gastrointestinal issues. The fruit contains an abundance of polyphenols, which contribute to its strong antioxidant properties. The antibacterial activity of fruit extracts derived from *P. emblica* against *Escherichia coli*, *Staphylococcus aureus*, and *Klebsiella pneumoniae* was determined along with the antibiotic-resistant variants extended-spectrum β-lactamase (ESBL) *E. coli*, methicillin-resistant *S. aureus* (MRSA), and ESBL *K. pneumoniae*. Disc diffusion and broth dilution assays were conducted to assess the activity of aqueous, methanolic, and ethyl acetate extracts, with large zones of inhibition of up to 15 mm on agar observed for *S. aureus* and MRSA. Minimum inhibitory concentration (MIC) values ranging from 158 to 1725 µg/mL were calculated. The aqueous and methanolic extracts of *P. emblica* were less active against *E. coli*, ESBL *E. coli*, *K. pneumoniae*, and *ESBL K. pneumoniae*, with the only noteworthy MIC (633 µg/mL) observed for the aqueous extract against *K. pneumoniae*. Interestingly, a lack of inhibition was observed on agar for any of the extracts against these bacteria. Liquid chromatography–mass spectrometry (LC-MS) analysis identified several notable flavonoids, phenolic acids, terpenoids, and tannins. Notably, *Artemia nauplii* bioassays indicated that all extracts were nontoxic. The antibacterial activity and absence of toxicity in *P. emblica* extracts suggest their potential as candidates for antibiotic development, highlighting the need for further mechanistic and phytochemical investigations.

## 1. Introduction

Antimicrobial resistance (AMR) poses a critical and growing challenge to global public health, resulting in heightened rates of illness, death, and economic hardship. Indeed, AMR resulted in approximately 1.27 million fatalities in 2019 [1]. The six primary pathogens associated with fatalities due to resistance include *Escherichia coli*, *Staphylococcus aureus*, *Klebsiella pneumoniae*, *Streptococcus pneumoniae*, *Acinetobacter baumannii*, and *Pseudomonas aeruginosa*, which together contributed to 929,000 deaths linked to AMR [2]. Multidrug-resistant strains encompass third-generation cephalosporin-resistant *E. coli*, fluoroquinolone-resistant *E. coli*, carbapenem-resistant *K. pneumoniae*, and third-generation cephalosporin-resistant *K. pneumoniae*, with each of these responsible for between 50,000 and 100,000 deaths attributable annually to their antibiotic resistance [2]. Additionally, the effectiveness of contemporary medical practices is compromised, increasing the risks associated with infections and procedures such as surgeries and cancer treatments. AMR also presents considerable economic implications, with projections indicating an extra $1 trillion required by healthcare to treat antibiotic-resistant infections by 2050, alongside a substantial effect on global GDP [1]. Combatting AMR necessitates the implementation of infection prevention strategies, ensuring access to diagnostics and treatments, and the pursuit of innovative solutions through extensive investigation. Notably, only less than 40 new antibacterial compounds are presently in the clinical trial stages [3]. It is important to note that the candidates aimed at World Health Organisation (WHO) priority pathogens are primarily derivatives of existing classes. Indeed, fewer than 25% of the drugs currently in the clinical development pipeline belong to a novel class or operate through a novel mechanism, and none of these show potential activity against Gram-negative WHO critical threat pathogens [3].

There is a growing interest in plant-derived antimicrobials, which offer distinctive bioactive compounds, extensive activity against various strains, and synergistic effects [4]. In contrast to synthetic antibiotics, numerous compounds present in medicinal plants, including flavonoids, terpenoids, and tannins, operate via mechanisms that diverge from conventional antibiotics, thereby complicating the ability of pathogens to rapidly develop resistance [4,5]. This alternative approach has the potential to reduce the spread of AMR by providing unique, multi-targeted pathways for bacterial inhibition, thereby lessening the rapid adaptation frequently observed with conventional drugs. Additionally, some medicinal plants can potentiate the action of clinical antibiotics, improving their effectiveness against resistant bacteria. Several plants demonstrate additive and/or synergistic effects when used alongside antibiotics, boosting antimicrobial potency and often enabling lower dosages, which can help minimise the selective pressure that drives resistance [4,6]. For instance, *Mangifera indica* Linn. ethanol extracts induce a fourfold reduction in the MIC of tetracycline and erythromycin when used in combination against *S. aureus* [7].

Recent studies reported significant antibacterial activity from a 70% ethanol extract of *Phyllanthus emblica* Linn. (*Emblica officinalis* Gaertn.) fruits and its phytochemicals (gallic acid, ellagic acid, rutin, and quercetin) against *S. aureus*, *E. coli*, and *K. pneumoniae*, with MIC values between 3.13 and 6.25 µg/mL [8,9]. That research also noted synergistic effects when combining the extract or compounds (gallic acid, ellagic acid, rutin, and quercetin) with ampicillin and chloramphenicol against the tested bacteria. However, the extracts and pure compounds were suspended in 10% dimethyl sulfoxide (DMSO) in that study. It is likely that this high DMSO concentration may inhibit bacterial growth, potentially producing falsely low MIC values. Indeed, standardised guidelines recommend that DMSO only be used in concentrations under 1% to prevent such interference [10]. Moreover, the extract volume used in the assays was not specified in that study, which may also compromise MIC accuracy. Another study reported synergistic interaction for 90% ethanol *P. emblica* extracts in combination with ciprofloxacin and azithromycin against *S. aureus* and *E. coli*, utilising disc diffusion assays [11]. The study demonstrated that the combination of extracts and antibiotics resulted in increased zone of inhibition (ZOI) values compared to their individual ZOI values. Notably, that study lacked important information regarding extract concentration, synergy testing protocols, and appropriate experimental controls, which constrains the reliability and reproducibility of the findings. Additionally, the final sample preparation for the disc diffusion assay used high concentrations of ethanol, which could impact bacterial growth and skew results, providing false positive results.

To address inconsistencies in the previous research on the antibacterial activities of *P. emblica* and the lack of comprehensive phytochemical analyses, we evaluated the antibacterial activity of aqueous, methanolic, and ethyl acetate fruit extracts against selected bacterial pathogens, including resistant strains of *S. aureus*, *E. coli*, and *K. pneumoniae.* Additionally, interactions between active extracts and reference antibiotics were investigated using fractional inhibitory concentration (FIC) assays to determine potential synergistic effects. Furthermore, LC-MS analysis was performed to characterise the phytochemical profiles of the extracts, identifying key flavonoids, phenolic acids, terpenoids, and tannins. Lastly, extract toxicity was assessed through *Artemia franciscana* nauplii lethality assays, providing preliminary understandings into their safety for therapeutic applications.

## 2. Materials and Methods

### 2.1. Plant Sources

The fruit powder of *Phyllanthus emblica* (batch no: AMP1020), manufactured by Aarshaveda, was purchased online from Sattvic (Melbourne, Australia). The supplier’s website was searched using the conventional Ayurvedic nomenclature of *P. emblica* (Amla, Indian gooseberry) to locate the product. The authenticity and purity of the plant material were verified by the provider, with the product sourced from India and the tree-ripened whole berries dried at room temperature and finely ground into the powdered form. Proper labelling was ensured, and voucher specimens (NBG-EO0220GU) for *P. emblica* were deposited at the School of Pharmacy and Medical Sciences, Griffith University (Southport, Australia).

### 2.2. Extract Preparation

The extracts were prepared using a previously developed methodology established by our group [12]. Briefly, one gram of *P. emblica* fruit powder was incubated in 50 mL of sterile deionised water (EO-Aq), methanol (EO-MeOH), or ethyl acetate (EO-EtOAc) for 24 h at room temperature with gentle oscillation. Organic solvents were of analytical grade and purchased from Thermo Fisher Scientific Inc. (Melbourne, Australia). The extracts were then vacuum filtered, and the aqueous extracts were dried using a laboratory freeze dryer (Martin Christ, Germany) for 72 h while organic extracts were evaporated at 42 °C. Dried extracts were weighed, reconstituted in 10 mL of 1% DMSO (Merck Life Science Pty. Ltd., Bayswater, Australia), and passed through 0.2 µm filters (Sarstedt Australia Pty. Ltd., Mawson Lakes, Australia), and stored at −20 °C.

### 2.3. Antibiotics and Bacterial Strains

Powdered and disc antibiotics, including penicillin G, ciprofloxacin, polymyxin B, oxacillin, amoxicillin, erythromycin, tetracycline, chloramphenicol, gentamicin, vancomycin, Augmentin^®^, and cefoxitin, were obtained from Merck Life Science Pty. Ltd. (Bayswater, Australia). Powdered antibiotics were prepared as 1 mg/mL stock solutions for broth microdilution and stored at −20 °C. Amoxicillin discs were prepared by applying 10 µL of a 10 µg/mL stock solution to sterile filter paper discs. Reference strains of *Klebsiella pneumoniae* (ATCC 13883), ESBL *K. pneumoniae* (ATCC 700603), *Escherichia coli* (ATCC 25922), *Staphylococcus aureus* (ATCC 25923), and MRSA (ATCC 43300) were sourced from American Type Culture Collection (ATCC) via In Vitro Technologies (Noble Park North, Australia), while a clinical isolate of ESBL *E. coli* was acquired from Gold Coast University Hospital (Southport, Australia). Mueller–Hinton (MH) agar and broth were used for bacterial cultures, with MRSA maintained at 35 °C [13] and other strains incubated at 37 °C for 18–24 h.

### 2.4. Antibacterial Susceptibility Screening

Antibacterial activities *of P. emblica* fruit extracts in MH agar were examined through a modified Kirby–Bauer disc diffusion method [12]. The abbreviations used in this study include EO-AQ (*P. emblica* aqueous extract), EO-MeOH (*P. emblica* methanol extract), and EO-EtOAc (*P. emblica* ethyl acetate extract). Reference antibiotics (Merck Life Sciences Pty. Ltd., Bayswater, Australia) are penicillin G (PEN G), erythromycin (ERY), tetracycline (TET), chloramphenicol (CHL), ciprofloxacin (CIP), polymyxin B (POL B), oxacillin (OXA), amoxicillin (AMX), gentamicin (GEN), vancomycin (VAN), Augmentin^®^ (AUG), and cefoxitin (CEF). Negative controls included 1% DMSO, while sterile water served as the blank.

### 2.5. Minimum Inhibitory Concentration (MIC)

The MIC values were assessed using a 96-well microdilution assay, where extracts and antibiotics were serially diluted and inoculated with bacterial suspensions [12]. Plates were incubated for 24 h at 37 °C (with the exception of MRSA, which was incubated at 35 °C), followed by p-iodonitrotetrazolium violet (Merck Life Sciences Pty. Ltd.) staining, which was employed to visualise bacterial inhibition. MIC was recorded as the lowest concentration preventing colour change. Activity was classified into six categories, ranging from inactive to highly active [14]. All experiments were performed in duplicate.

### 2.6. Assessment of Fractional Inhibitory Concentration (FIC)

Reference antibiotics and plant extracts with antibacterial activity were selected for combination studies at a 50:50 ratio to evaluate their interactions against susceptible bacterial pathogens. The interactions were evaluated by calculating the FIC for each component and determining the total FIC (ΣFIC) as described in previous methods [12]. Based on ΣFIC values, interactions were categorised as synergistic (≤0.5), additive (>0.5–1.0), indifferent (>1.0–4.0), or antagonistic (>4.0) [15].

### 2.7. Toxicity Evaluation

The toxicity of plant extracts was assessed following a previously described method using *Artemia franciscana* nauplii lethality assays (ALA) [12,16] with brine shrimp purchased from Aquabuy (Silverwater, Australia). Mortality was evaluated by counting live nauplii, and LC_50_ values were determined using probit analysis, representing the concentrations causing 50% lethality.

### 2.8. Non-Targeted Headspace Quantitative Analysis Using LC-MS

A comprehensive headspace metabolic profiling of all extracts was performed using a Vanquish Ultra High-Performance Liquid Chromatography (UHPLC) system coupled with an Orbitrap Exploris 120 mass spectrometer (Thermo Fisher, Melbourne, Australia). The system parameters, gradient flow, mobile phases, and data analysis of eluted compounds were conducted according to the methodology established by our group in previous research [12]. This methodology confirmed uniform gradient programming, optimal system configurations, and thorough examination of compound elution profiles, enabling precise and reproducible outcomes for the present study. Compound lists for each extract were exported to Excel (Version 2412), where potential compounds were identified by comparison. Duplicate entries were combined using pivot table analysis, and relative abundance was calculated as a percentage of the total peak area.

## 3. Results

### 3.1. Antibacterial Assays

Following extraction, the final concentrations of the *P. emblica* fruit extracts were 40.5, 47.3, and 6.9 mg/mL, respectively. The antibacterial efficacy of all extracts and control antibiotics was calculated using disc diffusion assays, quantified as ZOI values. Additionally, broth microdilution assays were used as a more quantitative assessment of antibacterial potency, reported as MIC values. The aqueous and methanol extracts demonstrated varying levels of antibacterial activity against *S. aureus*, *E. coli*, and *K. pneumoniae*, as well as their resistant strains, in both the disc diffusion and broth microdilution assays (Figure 1 and Table 1). The disc diffusion assays demonstrated that both the aqueous and methanolic extracts were active against *S. aureus* and MRSA, with inhibition zones measuring between 9.5 and 15 mm. In contrast, the ethyl acetate extract exhibited no activity against *S. aureus* and MRSA on agar. In broth microdilution assays, both aqueous and methanol extracts demonstrated good antibacterial activity against *S. aureus* and MRSA, with MIC values varying from 158 to 315 µg/mL. In contrast, the ethyl acetate extract demonstrated noteworthy to moderate activity against *S. aureus* and MRSA, with MIC values of 863 and 1725 µg/mL, respectively. The aqueous and methanol *P. emblica* extracts exhibited substantial antibacterial activity against MRSA, comparable to the efficacy of the conventional antibiotics penicillin G, amoxicillin, oxacillin, erythromycin, Augmentin^®^, and cefoxitin. This highlights the promise of plant-derived compounds to be effective alternatives or complementary agents in the fight against antibiotic-resistant infections.

None of the fruit extracts demonstrated antibacterial activity against *E. coli*, ESBL *E. coli*, *K. pneumoniae*, and ESBL *K. pneumoniae* in the disc diffusion assay. In contrast, the aqueous and methanol extracts exhibited low to noteworthy antibacterial activity against these pathogens in broth microdilution assays. The aqueous extracts showed low antibacterial activity against *E. coli*, ESBL *E. coli*, and ESBL *K. pneumoniae*, with MIC values of 5063 and 10,125 µg/mL, respectively. Notably, the aqueous extract displayed noteworthy antibacterial activity against *K. pneumoniae*, with an MIC value of 633 µg/mL. Furthermore, the methanolic extract exhibited low antibacterial activity against *E. coli*, *K. pneumoniae*, and ESBL *K. pneumoniae*, with a MIC of 2956 µg/mL. Interestingly, the methanol extract demonstrated moderate antibacterial activity against ESBL *E. coli*, with an MIC of 1478 µg/mL.

### 3.2. Combinatorial Studies: Fractional Inhibitory Concentration Determinations

Also examined were the interactions between combinations of *P. emblica* extracts and conventional antibiotics against the bacterial strains tested (Table 2). No synergistic interactions were detected for any combination in this study. Nine combinations exhibited additive effects, while thirty-eight combinations showed indifference. Furthermore, nine combinations exhibited antagonistic effects.

### 3.3. LC-MS Metabolomic Analysis and Compound Characterisation

Extract metabolomic profiles were acquired using LC-MS profiling. Compound identification was carried out by cross-referencing various databases whenever possible. The study sought to profile a diverse array of compounds, with a specific focus on flavonoids, tannins, and terpenoids. The majority of compounds in the extracts were of high polarity, as they were found to elute during the 30% to 90% acetonitrile gradient phase in the chromatogram (Appendix A) [17]. In contrast, organic acids and amines show a tendency to favour polar environments, leading to their earlier elution in the chromatogram. Conversely, lipophilic compounds that are of low polarity were eluted later in the gradient at elevated acetonitrile concentrations due to their enhanced affinity for the non-polar stationary phase. To create a thorough list of identified phytochemicals, only those compounds that corresponded with the data reported from at least one of the screened databases were incorporated (Appendix A). This study concentrated on phenolic acids, tannins, flavonoids, and terpenoids (Table 3) because of their significant biological properties [18]. These compounds are appreciated for their medicinal properties, including their antioxidant, anti-inflammatory, and antibacterial effects. Furthermore, their functions in plant defence underscore their promise as natural resources for drug development and therapeutic uses, preventing disease and increasing general health.

### 3.4. Toxicity Analysis

The toxicity of plant extracts was evaluated using *Artemia franciscana* lethality assays. The extracts were classified as toxic if they yielded LC_50_ values of 1000 µg/mL or lower following 24 h of exposure to the extracts [19]. The results for all extracts were comparable to the negative control (containing artificial seawater) (*p* > 0.05), as they did not cause ≥50% mortality at 1000 µg/mL. As such, they were deemed non-toxic.

## 4. Discussion

This study evaluated the antibacterial activity of *P. emblica* fruit extracts against both antibiotic-susceptible and resistant bacterial pathogens. Aqueous and methanolic extracts demonstrated antibacterial activity ranging from low to good, with methanolic extracts being the most potent on agar and in broth dilution assays. Ethyl acetate extracts showed moderate to notable activity, particularly against *S. aureus* and MRSA. The observed differences in antibacterial potency are likely attributable to variations in extract yields and phytochemical concentrations associated with solvent polarity. Methanol and water, as polar solvents, extract greater quantities of mid-to-high polarity phytochemicals [20], whereas ethyl acetate predominantly extracts mid-to-low polarity compounds. These variations in phytochemical composition influence antibacterial activity, as low-polarity or larger phytochemicals diffuse more slowly through solid agar, thereby affecting disc diffusion outcomes [21]. Additionally, the solubility of these phytochemicals in broth may influence the accuracy and consistency of MIC measurements [22]. Furthermore, prior research has highlighted that factors such as agar thickness and uniformity can significantly impact the size of inhibition zones in agar diffusion assays [23].

The MRSA strain demonstrated significant resistance to several widely used antibiotics, including β-lactams such as penicillin G, oxacillin, and amoxicillin, as well as macrolides including erythromycin. This resistance is further exacerbated by the rise of extended-spectrum β-lactamase (ESBL) enzymes, which render many β-lactam antibiotics less effective [24]. Similarly, macrolides, which are commonly used to inhibit bacterial protein synthesis, face challenges as MRSA exhibits resistance to these drugs. Such resistance substantially limits treatment options, emphasising the critical need for novel therapeutic approaches and drug candidates capable of bypassing or overcoming these resistance mechanisms. In this context, identifying innovative compounds with unique antibacterial properties becomes increasingly important. Interestingly, all extracts from *P. emblica* exhibited good to moderate antibacterial activity against *S. aureus* and MRSA, with MIC ranging from 158 µg/mL to 1725 µg/mL. These results suggest that the MRSA resistance mechanisms may have minimal impact on the efficacy of the active compounds within these extracts. This could imply that the compounds either target distinct bacterial pathways or actively inhibit the mechanisms that confer antibiotic resistance. These findings underscore the potential of such plant-based extracts as promising candidates for addressing the growing challenge of antibiotic resistance.

The *mecA* gene is a pivotal factor in conferring resistance to MRSA, as it encodes a unique penicillin-binding protein, PBP2a, which exhibits a reduced affinity for β-lactam antibiotics [25]. This protein allows MRSA to continue synthesising its cell walls even in the presence of β-lactam antibiotics, rendering these medications largely ineffective. As a result, the mode of action of phytochemicals present in the extracts may differ significantly from that of β-lactam antibiotics, even in strains resistant to these drugs. Alternatively, the phytochemicals in these extracts could interfere with the bacterial defence mechanisms, effectively sensitising the bacteria to the antibiotics and enhancing their potency [25]. This finding is particularly noteworthy, as the MRSA strain examined in this study displayed substantially higher levels of resistance (when compared to the susceptible strain) to a broad spectrum of antibiotics, including the β-lactam, macrolide, and fluoroquinolone classes. Such observations highlight the potential of these phytochemical-rich extracts as a promising avenue for overcoming antibiotic resistance and addressing the growing challenges posed by resistant bacterial strains.

The aqueous *P. emblica* extracts exhibited low antibacterial activity against *E. coli* and ESBL *E. coli*, with a MIC of 5063 µg/mL. In contrast, methanol extracts demonstrated better antibacterial efficacies against both pathogens, with MIC values of 2956 µg/mL and 1478 µg/mL, respectively. Similarly, methanol extracts exhibited the same MIC of 2956 µg/mL against *K. pneumoniae* and ESBL *K. pneumoniae*. Interestingly, the aqueous extract showed noteworthy antibacterial activity against *K. pneumoniae*, with an MIC of 633 µg/mL. These findings indicate that aqueous and methanol extracts may contain bioactive compounds with broad-spectrum activity against *E. coli* and *K. pneumoniae*, including resistant strains. This efficacy may result from mechanisms of action distinct from those of β-lactam antibiotics. For instance, the extracts might interfere with bacterial cell wall formation, membrane functionality, or other vital processes unrelated to β-lactam mechanisms [26]. Furthermore, the antibacterial activity of these extracts may not directly inhibit ESBL enzymes but could operate through alternative pathways. Further investigation is required to determine whether the extracts directly disrupt resistance in the ESBL-producing pathogens or act via other, unidentified antibiotic mechanisms. Such studies should assess the inhibitory effects of the extracts or isolated components on β-lactamase activity and evaluate their influence on the expression of ESBL resistance genes.

Our study also investigated the potential of combining *P. emblica* extracts with conventional antibiotics. This approach shows significant promise for the development of novel antibiotic therapies, particularly as bacterial resistance to conventional antibiotics continues to rise. Plant-derived compounds may offer unique mechanisms to inhibit or block these resistance pathways, enhancing the efficacy of existing antibiotics [27]. Our objective was to enhance antibiotic efficacy and counteract bacterial resistance mechanisms by combining them with plant extracts. A well-known example of this approach is Augmentin^®^, a formulation that contains amoxicillin and clavulanate, which improves treatment outcomes by addressing resistance [28]. The clavulanate inhibits β-lactamase enzymes that are present within resistant species, allowing amoxicillin to be effective. It achieves this by permanently binding to the active site of the enzyme, preventing antibiotic destruction.

Our study observed additive interactions between the antibiotics and plant extracts, including penicillin G, tetracycline, ciprofloxacin, and vancomycin, against *S. aureus* and MRSA. Additive interactions suggest that the overall impact of the plant extracts and antibiotics is equivalent to the total of their separate effects. Although this does not indicate a synergistic enhancement, it illustrates that the plant extracts offer complementary antibacterial support in conjunction with the antibiotics, and therefore the use in combination may be beneficial.

The additive effect observed may originate from the plant extracts interacting with bacterial pathways or mechanisms that are different from those influenced by the antibiotics. For instance, penicillin G and vancomycin target bacterial cell wall synthesis, while plant extracts may interfere with bacterial membranes or intracellular processes [29], thereby increasing the stress on the bacterial cells. Similarly, tetracycline inhibits protein synthesis, and ciprofloxacin interferes with DNA replication, but the plant extracts might simultaneously target metabolic pathways or reduce efflux pump activity [29], thereby contributing to the overall antibacterial effect. Furthermore, the methanol *P. emblica* extract demonstrated an additive interaction with tetracycline in combating *K. pneumoniae*, likely due to their interference with tetracycline-specific efflux pumps [30], which is a key mechanism of resistance. Inhibiting these pumps may enable the extracts to prolong the retention of tetracycline within bacterial cells, thereby increasing its efficacy. Whilst ribosomal modifications may also play a role in tetracycline resistance, this mechanism is not as common, indicating that the extracts may primarily focus on inhibiting efflux pump activity to enhance the antibacterial effects of tetracycline.

These findings underscore the potential of integrating plant-derived compounds [29] with conventional antibiotics to combat bacterial resistance. Such combinations could be particularly valuable in addressing multidrug-resistant pathogens, including MRSA. Further research to elucidate the mechanisms underlying these interactions is essential to optimise their application in clinical settings and develop innovative therapeutic approaches.

Notably, the polymyxin B and *P. emblica* extract combination exhibited substantial antagonistic effects against *E. coli*, ESBL *E. coli*, *K. pneumoniae*, and ESBL *K. pneumoniae*. This may be influenced by pH variations in the broth, as polymyxin B’s membrane-disrupting activity is highly pH-sensitive, since its efficacy is suppressed under alkaline and acidic conditions [31]. The presence of plant extracts could alter the pH, potentially diminishing the antibacterial potency of both polymyxin B and the phytochemicals. Furthermore, bioactive phytochemicals may bind to polymyxin B, impairing its absorption and effectiveness [32]. Understanding these dynamics is essential for optimising combination therapies, and future studies will be critical to explore these interactions in greater detail.

LC-MS metabolomics analysis of the *P. emblica* extracts showed the presence of multiple phytochemicals, including flavonoids, tannins, terpenoids, and phenolic acids (Table 3). Notable phytochemicals identified in the *P. emblica* extracts include chebulic acid (Figure 2A), fukiic acid (Figure 2B), citric acid (Figure 2C), 6-galloylglucose (Figure 2D), pyrogallol-2-O-glucuronide (Figure 2E), pyrogallol (Figure 2F), phloroglucinol (Figure 2G), 1,2,6-tri-O-galloyl-β-D-glucopyranose (Figure 2H), ellagic acid (Figure 2I), gallic acid (Figure 2J), kaempferol (Figure 2K), and quercetin (Figure 2L). Previous studies also identified other various phytochemicals in the fruit extracts of *P. emblica*, including ellagic acid, gallic acid, ascorbic acid, chebulic acid, chebulinic acid, kaempferol, quercetin, corilagin, emblicanin, and pedunculagin [13,33,34]. Our previous study conducted GC-MS analysis on the water, methanol, and ethyl acetate fruit extracts of *P. emblica* and reported the presence of 2-butoxy-ethanol, octanal, 2-ethyl-1-hexanol, nonanal, methoxycitronellal, endo-borneol, terpinen-4-ol, 2-methyl-decane, carvone, 2-phenylbutanal, 2,2,4-trimethyl-1,3-pentanediol diisobutyrate, 2-ethyl-3-hydroxyhexyl 2-methylpropanoate and 1-isobutyl 4-isopropyl 3-isopropyl-2,2-dimethylsuccinate [19].

Our study identified chebulic acid in all extracts of *P. emblica*. Studies in the literature are lacking in terms of the antibacterial potential of chebulic acid. However, previous studies reported that chebulic acid showed significant protection against endothelial cell dysfunction and has antioxidant properties [35,36]. Our study also identified fukiic acid, a polyphenol in the ethyl acetate extracts of *P. emblica*. Currently, there is limited scientific literature specifically addressing the antibacterial properties of fukiic acid.

The fruit extracts of *P. emblica* were found to be relatively abundant in citric acid and isocitric acid. Citric acid has demonstrated moderate antimicrobial activity against *E. coli* and *S. aureus*. Studies have reported MICs of 60 mg/mL for *E. coli* and *S. aureus* [37]. Another study noted MICs of >1000 µg/mL for *S. aureus* and 500 µg/mL for *E. coli* [38]. The antimicrobial efficacy of citric acid is influenced by pH, as it exhibits greater activity at low pH levels, particularly between its first and second pKa values (3.1 and 4.7) [39]. At these pH levels, citric acid remains largely undissociated, enhancing its ability to penetrate microbial membranes. A previous study demonstrated that citric acid induces antibiotic tolerance in bacteria through significant alterations in metabolism and oxidative stress [40]. Specifically, citric acid activates the glyoxylate cycle, suppresses the tricarboxylic acid (TCA) cycle, reduces ATP production, and mitigates oxidative stress, collectively impairing the efficacy of antibiotics [40]. The findings advocate for a balanced approach to citric acid usage, especially in medical and dietary contexts, to prevent compromising antibiotic effectiveness.

We have also identified galloyl-glucose derivatives in the *P. emblica* fruit extracts, including 6-galloylglucose, methyl 6-O-galloyl-β-D-glucopyranoside, 1,6-bis-O-(3,4,5-trihydroxybenzoyl) hexopyranose, methyl 4,6-di-O-galloyl-β-D-glucopyranoside, 1,2,6-trigalloyl-β-D-glucopyranose, 1,3,4-trigalloyl-β-D-glucopyranose, and 2-cinnamoyl-1,6-digalloyl-β-D-glucopyranose. Previous studies also identified several galloyl-glucose derivatives in the *P. emblica* fruit extracts [13,33,34,41,42]. Additionally, efflux pump inhibitory activity has been reported for 1,2,6-tri-O-galloyl-β-d-glucopyranose against MDR uropathogenic *E. coli* [43]. That compound showed efflux pump inhibition, which was confirmed by ethidium bromide accumulation and efflux assays, suggesting this mechanism may contribute to its antibacterial effects. Furthermore, this compound exhibited the MIC of 10–16 µg/mL (15.72–25.15 µM) against *E. coli* strains and demonstrated synergistic antibiofilm activity in combination with gentamicin and trimethoprim [44].

Our study also identified some other notable phytochemicals in the *P. emblica* extracts, including gallic acid, pyrogallol, ellagic acid, kaempferol, and quercetin. Previous studies have reported that gallic acid and pyrogallol have notable synergistic effects when combined with conventional antibiotics against *Staphylococcus aureus*. Gallic acid reduced the MIC of norfloxacin from 156.3 μg/mL to 49.21 μg/mL and gentamicin from 49.21 μg/mL to 2.44 μg/mL, indicating substantially enhanced antibacterial efficacy [45]. Similarly, pyrogallol showed significant synergy, reducing the MIC of norfloxacin by 49.98% (from 156.3 μg/mL to 78.13 μg/mL) and achieving an even greater effect with gentamicin, reducing the MIC from 49.21 μg/mL to 2.44 μg/mL. Among the tested combinations, gentamicin and pyrogallol yielded the most substantial reduction, highlighting its potential as an effective therapeutic strategy against *S. aureus*. In addition, pyrogallol also exhibited notable antimicrobial activity against methicillin-susceptible *S. aureus*, methicillin-resistant *S. aureus*, *E. coli*, colistin-resistant *E. coli*, and colistin-resistant *K. pneumoniae*, with an MIC of 250 μg/mL and a minimal bactericidal concentration (MBC) of 250–500 μg/mL [46]. Ellagic acid reduced the MIC of tetracycline, chloramphenicol, and tobramycin by up to fourfold against MDR *E. coli* [47].

Kaempferol exhibits significant antibacterial activity, particularly when combined with colistin, showing synergistic effects against colistin-resistant Gram-negative bacteria such as *K. pneumoniae* and *E. coli*. For instance, kaempferol reduced MICs of colistin by up to fourfold in resistant strains [48]. Additionally, kaempferol disrupts bacterial cell walls and biofilm formation, highlighting its potential as an adjunctive agent in combating antibiotic resistance. Quercetin exhibits notable antibacterial effects against *S. aureus* and *E. coli*, with MICs of 20 and 500 μg/mL, respectively [49]. Additionally, quercetin exhibited synergistic interactions with gentamicin and ceftriaxone against MDR *E. coli* [50].

Toxicity assessments using *Artemia nauplii* confirmed that all fruit extracts of *P. emblica* are non-toxic, suggesting their potential safety as antimicrobial agents. However, further evaluation using diverse mammalian cell lines is necessary to confirm their suitability for medical applications. Overall, this study highlights fruit extracts of *P. emblica* as promising sources of antimicrobial compounds for future research and combating bacterial infections.

## 5. Conclusions

The growing prevalence of antibiotic-resistant bacteria highlights the urgent need for novel antibacterial agents, with natural products emerging as promising candidates. Our study demonstrated that fruit extracts of *P. emblica* effectively inhibit the growth of both resistant and susceptible bacterial strains. Furthermore, the extracts additively enhanced the efficacy of conventional antibiotics, including penicillin G, ciprofloxacin, vancomycin, and tetracycline, potentially restoring their activity against resistant bacteria. This potentiation may be attributed to the extracts’ ability to deactivate bacterial β-lactamase enzymes and efflux pumps, thereby increasing intracellular antibiotic concentrations. Several phytochemicals identified in the extracts likely contribute to these effects, making them valuable targets for the development of new antibacterial agents. Future research should focus on confirming the potentiating mechanisms of phytochemicals and exploring additional pathways. In particular, studies evaluating the extracts for the ability to inhibit β-lactamase enzymes and efflux mechanisms are required. Additionally, qPCR studies would be useful to determine if the expression of the enzymes and efflux proteins is affected by the extract components.

## Figures and Tables

**Figure 1 microorganisms-13-00611-f001:**
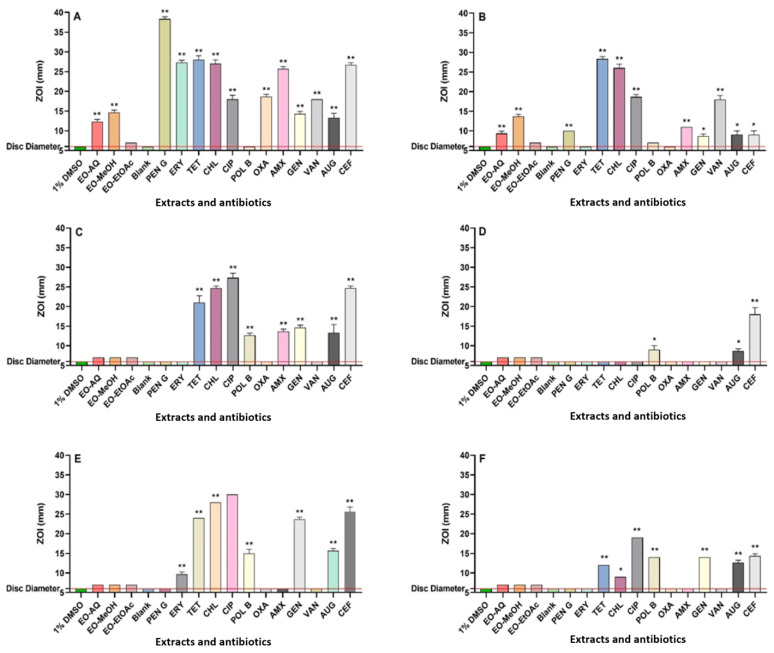
Antimicrobial activities of *P. emblica* fruit extracts were assessed using disc diffusion assays against the following: (**A**) *S. aureus*, (**B**) MRSA, (**C**) *E. coli*, (**D**) ESBL *E. coli*, (**E**) *K. pneumoniae*, and (**F**) ESBL *K. pneumoniae*. The horizontal red line at 6 mm on the *y*-axis marks the disc diameter used in the assay. A single asterisk (*) indicates *p*-values < 0.01, while a double asterisk (**) represents *p*-values < 0.001.

**Figure 2 microorganisms-13-00611-f002:**
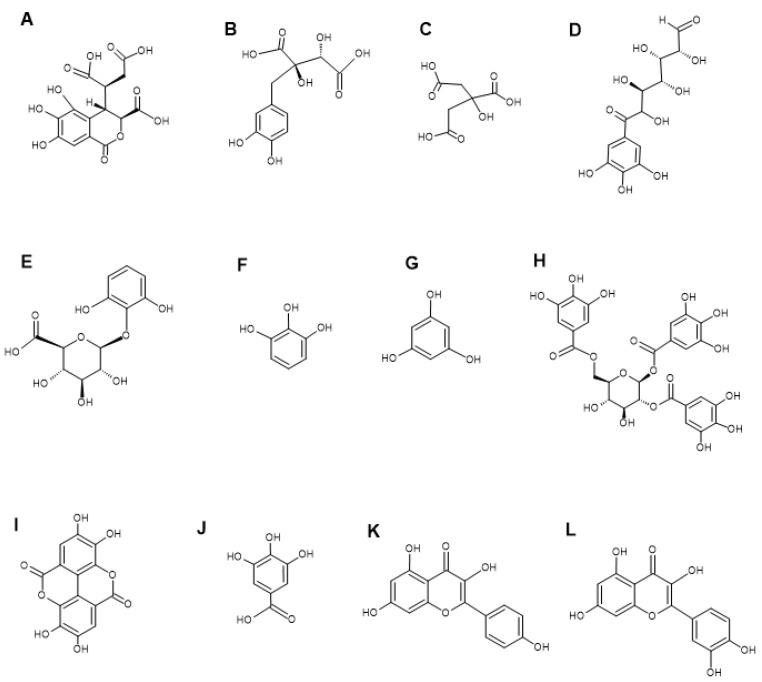
Structures of notable compounds identified in the fruit extracts of *P. emblica*. Chebulic acid (**A**), fukiic acid (**B**), citric acid (**C**), 6-galloylglucose (**D**), pyrogallol-2-O-glucuronide (**E**), pyrogallol (**F**), phloroglucinol (**G**), 1,2,6-tri-O-galloyl-β-D-glucopyranose (**H**), ellagic acid (**I**), gallic acid (**J**), kaempferol (**K**), and quercetin (**L**). The Chemsketch software (version 2023.2.4) was used to make the chemical structures.

**Table 1 microorganisms-13-00611-t001:** Minimum inhibitory concentration (MIC) values (µg/mL) for aqueous (EO-AQ), methanol (EO-MeOH), and ethyl acetate (EO-EtOAc) fruit extracts of *P. emblica*, and reference antibiotics as positive controls, against six bacterial species.

Extract Type or Antibiotic	Bacterial Species and MIC (µg/mL)					
	*S. aureus*	MRSA	*E. coli*	ESBL *E. coli*	*K. pneumoniae*	ESBL *K. pneumoniae*
EO-AQ	316	158	**5063**	**5063**	633	10,125
EO-MeOH	185	185	**2956**	1478	**2956**	**2956**
EO-EtOAc	863	1725	>10,000	>10,000	>10,000	>10,000
PENG	**1.25**	-	-	-	-	-
ERY	**0.31**	-	-	-	-	-
TET	**0.16**	**0.04**	**0.31**	-	**0.625**	-
CHL	-	-	**2.5**	-	**2.5**	-
CIP	**0.16**	**0.625**	**0.02**	-	**0.02**	**0.16**
POLB	-	-	**0.02**	**0.02**	**0.02**	**0.04**
OXA	**0.16**	-	-	-	-	-
AMX	**0.625**	-	-	-	-	-
GEN	-	-	**0.625**	-	**0.625**	-
VAN	**1.25**	**1.25**	-	-	-	-

Extract MIC values were categorised as follows: inactive (MIC > 10,000 µg/mL), low activity (2000–5000 µg/mL, **bold**), moderate activity (1000–2000 µg/mL, blue), noteworthy activity (400–1000 µg/mL, red), and good activity (100–400 µg/mL, green). Active antibiotics are highlighted in **bold**, while those with MIC > 2.5 µg/mL are considered inactive and denoted as (-).

**Table 2 microorganisms-13-00611-t002:** ∑FIC values for combinations of *P. emblica* fruit extracts and conventional antibiotics.

Bacteria	Extracts	Antibiotics									
		PENG	ERY	TET	CHL	CIP	POLB	OXA	AMX	GEN	VAN
*S. aureus*	EO-AQ	0.53	1.25	0.75	-	1.50	-	1.50	1.13	-	1.06
	EO-MeOH	0.53	1.13	1.25	-	1.25	-	1.25	1.06	-	1.03
	EO-EtOAc	1.00	1.25	1.13	-	2.15	-	2.25	1.50	-	2.00
MRSA	EO-AQ	**-**	**-**	1.00	-	1.06	**-**	**-**	**-**	**-**	1.03
	EO-MeOH	**-**	**-**	1.00	-	1.06	**-**	**-**	**-**	**-**	1.03
	EO-EtOAc	**-**	**-**	1.02	-	0.63	**-**	**-**	**-**	**-**	0.75
*E. coli*	EO-AQ	**-**	**-**	1.25	1.50	8.31	63.50	**-**	**-**	3.00	**-**
	EO-MeOH	**-**	**-**	1.50	1.25	2.25	64.50	**-**	**-**	2.00	**-**
ESBL *E. coli*	EO-AQ	**-**	**-**	**-**	**-**	**-**	63.50	**-**	**-**	**-**	**-**
	EO-MeOH	**-**	**-**	**-**	**-**	**-**	16.50	**-**	**-**	**-**	**-**
*K. pneumoniae*	EO-AQ	**-**	**-**	1.25	2.12	3.00	17.50	**-**	**-**	2.50	**-**
	EO-MeOH	**-**	**-**	1.00	1.25	2.50	32.25	**-**	**-**	2.00	**-**
ESBL *K. pneumoniae*	EO-AQ	**-**	**-**	**-**	**-**	2.13	31.75	**-**	**-**	**-**	**-**
	EO-MeOH	**-**	**-**	**-**	**-**	2.50	8.25	**-**	**-**	**-**	**-**

∑FIC values of aqueous (EO-AQ), methanol (EO-MeOH), and ethyl acetate (EO-EtOAc) extracts of *P. emblica* in combination with reference antibiotics against *S. aureus*, MRSA, *E.* coli, ESBL *E. coli*, *K. pneumoniae*, and ESBL *K. pneumoniae*. Additive interaction: >0.5 ≤ 1.00 (blue color), indifferent interaction: >1.01–≤4.00, antagonistic interaction: >4.0 (red color). - indicates the extract and/or the antibiotic was inactive against the bacteria being tested.

**Table 3 microorganisms-13-00611-t003:** LC-MS analysis using negative ionisation mode showing the putative identification and % relative abundance of phytochemicals in the fruit extracts of *P. emblica*. Phytochemicals absent in any extract are indicated as (-).

Retention Time (min)	Molecular Mass	Empirical Formula	Putative Compounds	% Relative Abundance		
				AQ	MeOH	EtOAc
1.191	148.037	C_5_ H_8_ O_5_	δ-Ribono-1,4-lactone	-	1.51%	1.68%
1.251	160.0369	C_6_ H_8_ O_5_	Cortalcerone	-	1.14%	-
1.269	116.0108	C_4_ H_4_ O_4_	Maleic acid	0.75%	0.97%	-
1.371	194.0789	C_7_ H_14_ O_6_	Methyl β-D-glucopyranoside	-	0.09%	-
1.373	356.0377	C_14_ H_12_ O_11_	**(+)-Chebulic acid**	0.15%	0.95%	2.55%
1.377	158.02104	C_6_ H_6_ O_5_	2-Methylene-4-oxopentanedioic acid	-	-	2.37%
1.41	100.0523	C_5_ H_8_ O_2_	Tiglic acid	-	0.50%	-
1.454	137.0475	C_7_ H_7_ N O_2_	Trigonelline	2.53%	0.03%	-
1.457	130.02623	C_5_ H_6_ O_4_	Mesaconic acid	0.48%	-	-
1.464	104.01077	C_3_ H_4_ O_4_	Malonic acid	0.23%	-	-
1.589	206.0424	C_7_ H_10_ O_7_	2-Methylcitric acid	-	0.06%	-
1.593	96.02098	C_5_ H_4_ O_2_	2-Furancarboxaldehyde	0.06%	0.07%	0.38%
1.597	272.05292	C_11_ H_12_ O_8_	Fukiic acid	-	-	15.32%
1.619	85.089	C_5_ H_11_ N	Piperidine	0.06%	0.34%	0.12%
1.752	210.03735	C_6_ H_10_ O_8_	D-Saccharic acid	0.97%	-	8.83%
1.769	192.0269	C_6_ H_8_ O_7_	Isocitric acid	15.42%	17.26%	1.61%
1.814	192.02685	C_6_ H_8_ O_7_	Citric acid	5.62%	-	2.61%
1.933	370.0534	C_15_ H_14_ O_11_	2-O-Caffeoylhydroxycitric acid	-	0.16%	-
1.947	206.02141	C_10_ H_6_ O_5_	**Flaviolin**	-	-	1.08%
1.968	316.0794	C_13_ H_16_ O_9_	Ginnalin B	-	0.03%	-
2.066	134.0215	C_4_ H_6_ O_5_	D-(+)-Malic acid	0.38%	2.96%	2.41%
2.067	148.03702	C_5_ H_8_ O_5_	Ribonolactone	0.09%	-	-
2.084	634.08037	C_27_ H_22_ O_18_	**Corilagin**	0.30%	-	-
2.165	222.05273	C_11_ H_10_ O_5_	Isofraxidin	0.01%	-	-
2.186	178.02655	C_9_ H_6_ O_4_	Aesculetin	0.01%	-	0.03%
2.303	332.0742	C_13_ H_16_ O_10_	**6-Galloylglucose**	19.79%	2.80%	7.03%
2.375	204.00925	C_7_ H_8_ O_5_ S	O-methoxy catechol-O-sulphate	0.01%	-	-
2.631	240.06348	C_11_ H_12_ O_6_	Lignicol	0.04%	-	-
2.786	346.0902	C_14_ H_18_ O_10_	**Methyl 6-O-galloyl-β-D-glucopyranoside**	0.01%	0.01%	-
3.272	342.09498	C_15_ H_18_ O_9_	Glucocaffeic acid	-	-	2.02%
3.418	302.0638	C_12_ H_14_ O_9_	**Pyrogallol-2-O-glucuronide**	10.94%	4.52%	1.03%
3.48	296.05274	C_13_ H_12_ O_8_	Caffeoylmalic acid	0.01%	-	-
3.914	484.0849	C_20_ H_20_ O_14_	**Hamamelitannin**	0.35%	0.74%	0.11%
3.919	126.0316	C_6_ H_6_ O_3_	Phloroglucinol	1.55%	0.76%	0.79%
4.151	232.00061	C_12_ H_8_ O S_2_	Arctinal	-	-	0.01%
4.236	348.08412	C_17_ H_16_ O_8_	3,5,7,3′,4′,5′-Hexahydroxy-6,8-dimethylflavanone	-	-	1.01%
4.585	312.0479	C_13_ H_12_ O_9_	Caftaric acid	-	1.69%	-
4.679	484.0857	C_20_ H_20_ O_14_	1,6-bis-O-(3,4,5-trihydroxybenzoyl) hexopyranose	1.59%	7.35%	3.59%
6.312	264.027	C_13_ H_12_ O_2_ S_2_	Arctinol	T	0.01%	-
8.821	634.0809	C_27_ H_22_ O_18_	**Sanguiin H4**	-	0.03%	1.77%
9.699	126.0317	C_6_ H_6_ O_3_	**Pyrogallol**	-	9.18%	6.74%
9.898	498.101	C_21_ H_22_ O_14_	**Methyl 4,6-di-O-galloyl-β-D-glucopyranoside**	-	0.01%	-
10.264	314.0635	C_13_ H_14_ O_9_	β-D-glucopyranuronic acid	0.99%	2.18%	-
10.44	636.0962	C_27_ H_24_ O_18_	**1,2,6-trigalloyl-β-D-glucopyranose**	-	1.30%	-
10.872	434.04837	C_19_ H_14_ O_12_	**Ellagic acid arabinoside**	-	-	1.05%
10.925	176.0319	C_6_ H_8_ O_6_	Ascorbic acid	-	0.11%	-
10.932	636.09613	C_27_ H_24_ O_18_	1,3,4-trigalloyl-β-D-glucopyranose	0.02%	-	0.16%
11.195	170.0214	C_7_ H_6_ O_5_	**Gallic acid**	1.19%	0.38%	1.42%
11.221	442.09013	C_22_ H_18_ O_10_	**Robinetinidol 3-O-gallate**	-	-	0.01%
11.422	464.0953	C_21_ H_20_ O_12_	**Myricitrin**	0.14%	0.21%	-
11.608	180.09368	C_14_ H_12_	Stilbene	-	-	2.04%
11.616	172.0735	C_8_ H_12_ O_4_	(-)-Corey lactone	0.03%	0.06%	-
11.641	448.0641	C_20_ H_16_ O_12_	**Ellagic acid 2-rhamnoside**	0.01%	0.01%	1.06%
11.663	478.11122	C_22_ H_22_ O_12_	**6-Methoxyluteolin 7-glucoside**	0.01%	-	-
11.731	600.1113	C_28_ H_24_ O_15_	**Isoorientin 2″-O-gallate**	0.01%	0.01%	-
11.746	162.06799	C_10_ H_10_ O_2_	4,5-dihydro-1-benzoxepin-3(2H)-one	-	-	0.01%
11.916	448.1005	C_21_ H_20_ O_11_	**Trifolin**	0.59%	0.55%	0.31%
11.917	286.0476	C_15_ H_10_ O_6_	**Kaempferol**	-	0.46%	0.14%
11.963	184.037	C_8_ H_8_ O_5_	**Methyl gallate**	0.24%	0.90%	-
11.965	170.0214	C_7_ H_6_ O_5_	2,4,6-trihydroxybenzoic acid	0.24%	1.69%	1.27%
11.978	310.1049	C_15_ H_18_ O_7_	(E)-1-O-cinnamoyl-β-D-glucose	-	3.86%	2.01%
11.99	302.0062	C_14_ H_6_ O_8_	**Ellagic acid**	12.37%	7.84%	9.17%
11.991	302.0423	C_15_ H_10_ O_7_	**Quercetin**	0.18%	0.27%	-
12.033	594.101	C_29_ H_22_ O_14_	**Epicatechin 3,5-di-O-gallate**	-	T	-
12.179	272.0683	C_15_ H_12_ O_5_	**Naringeninchalcone**	0.04%	0.06%	-
12.292	492.0908	C_22_ H_20_ O_13_	**6-Methoxyluteolin 7-glucuronide**	0.01%	0.03%	1.01%
12.307	286.04732	C_15_ H_10_ O_6_	**Fisetin**	0.31%	-	-
12.324	148.0524	C_9_ H_8_ O_2_	Cinnamic acid	1.01%	1.26%	1.45%
12.437	349.19973	C_18_ H_27_ N_3_ O_4_	Coutaric acid	0.02%	-	-
12.464	474.07948	C_22_ H_18_ O_12_	Chicoric acid	0.01%	-	-
12.48	292.09429	C_15_ H_16_ O_6_	(S)-Angelicain	0.01%	-	-
12.545	432.1059	C_21_ H_20_ O_10_	**Afzelin**	-	0.03%	-
12.674	550.1689	C_26_ H_30_ O_13_	Licuroside	-	0.01%	0.01%
12.757	150.06792	C_9_ H_10_ O_2_	Hydrocinnamic acid	-	-	0.04%
12.919	185.14141	C_10_ H_19_ N O_2_	1-Methylpiperidin-4-yl butanoate	-	-	0.09%
12.937	304.0581	C_15_ H_12_ O_7_	Nigrescin	0.02%	0.02%	-
12.983	572.1901	C_29_ H_32_ O_12_	Amorphigenin O-glucoside	0.02%	0.02%	-
12.996	262.01149	C_13_ H_10_ O_2_ S_2_	Arctinone A	0.01%	-	-
13.009	262.0477	C_13_ H_10_ O_6_	Maclurin	-	T	-
13.026	444.1054	C_22_ H_20_ O_10_	**3′-O-Methylderhamnosylmaysin**	0.08%	0.06%	0.01%
13.043	524.15321	C_24_ H_28_ O_13_	**Barbatoflavan**	-	-	1.01%
13.046	538.16926	C_25_ H_30_ O_13_	Lippioside I	T	-	-
13.058	218.05794	C_12_ H_10_ O_4_	Liqcoumarin	T	-	-
13.159	506.1062	C_23_ H_22_ O_13_	**Quercetin 3- (6″-ethylglucuronide)**	-	0.04%	-
13.188	504.34447	C_30_ H_48_ O_6_	Protobassic acid	-	-	1.02%
13.188	486.334	C_30_ H_46_ O_5_	Bassic acid	-	-	1.01%
13.561	400.1155	C_21_ H_20_ O_8_	**Torosaflavone A**	0.02%	0.03%	-
13.647	462.1162	C_22_ H_22_ O_11_	Leptosin	T	0.03%	0.01%
13.722	274.08384	C_15_ H_14_ O_5_	Phloretin	0.02%	-	-
13.874	614.1273	C_29_ H_26_ O_15_	**2-Cinnamoyl-1,6-digalloyl-β-D-glucopyranose**	-	0.01%	-
13.92	260.15199	C_15_ H_20_ N_2_ O_2_	Baptifoline	T	-	-
14.278	148.0523	C_9_ H_8_ O_2_	*p*-Coumaraldehyde	0.18%	0.26%	-
14.472	290.0788	C_15_ H_14_ O_6_	Marshrin	-	0.04%	-
14.812	286.0477	C_15_ H_10_ O_6_	Maritimetin	-	0.08%	-
14.829	580.1582	C_30_ H_28_ O_12_	**Naringenin 7- (2-p-Coumaroylglucoside)**	0.08%	-	-
16.15	316.13099	C_18_ H_20_ O_5_	**4,2′,6′-Trihydroxy-4′-methoxy-3′,5′-dimethyldihydrochalcone**	-	-	1.01%
17.041	184.09993	C_12_ H_12_ N_2_	Harmalan	-	-	0.02%
17.512	286.12056	C_17_ H_18_ O_4_	**4-Hydroxy-2′,4′-dimethoxydihydrochalcone**	T	-	-
17.969	256.07342	C_15_ H_12_ O_4_	Isoliquiritigenin	-	-	1.01%
18.036	314.11518	C_18_ H_18_ O_5_	4′-Hydroxyenterolactone	-	-	0.06%
18.608	242.09407	C_15_ H_14_ O_3_	2′,4′-Dihydroxydihydrochalcone	-	-	0.03%
18.835	388.20942	C_19_ H_32_ O_8_	5a,6a-Epoxy-7E-megastigmene-3a,9e-diol 3-glucoside	-	-	0.08%
19.447	226.099	C_15_ H_14_ O_2_	**7-Hydroxyflavan**	-	0.03%	0.02%

Compounds identified in the fruit extracts of *P. emblica* with relative abundances below 0.01% of the total area were considered trace amounts and labelled as ‘T’. Tannin components are indicated with **blue** text, whilst flavonoids are indicated in **red**. Components with both tannin and flavonoid moieties are highlighted in **green**. AQ = aqueous extracts, MeOH = methanolic extracts, EtOAc = ethyl acetate extracts.

## Data Availability

The original contributions presented in this study are included in the article/Appendix A. Further inquiries can be directed to the corresponding author.

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
