# Peer review of "Phyllanthus emblica*: Phytochemistry, Antimicrobial Potential with Antibiotic Enhancement, and Toxicity Insights"

_microorganisms, 2025, doi:10.3390/microorganisms13030611_

Round 1
Reviewer 1 Report
Comments and Suggestions for Authors
The authors examined the antibacterial activity of fruit extracts of P. emblica against Staphylococcus aureus, Escherichia coli, and Klebsiella pneumoniae, along with the antibiotic-resistant variants methicillin-resistant S. aureus (MRSA), extended-spectrum β-lactamase (ESBL) E. coli, and ESBL K. pneumoniae. Study results demonstrated that fruit extracts of P. emblica inhibited the growth of resistant and susceptible bacterial strains and enhanced the efficacy of conventional antibiotics. This study suggests that P. emblica extracts should be considered candidates for antibiotic development, highlighting the need for further mechanistic and phytochemical investigations. The study is interesting, scientifically important, and deserves attention.
Comments:
- Please correct the typo error in line 156- „strains. “
- It is unclear why the authors use the ALA test for toxicity assessment. ALA test is suitable for assessing toxicity from an environmental perspective, but direct applicability to human health risk assessment is limited. The authors should consider excluding that analysis from the manuscript.
-Lines 314-316. According to the authors' criteria, MIC values from 158 μg/mL to 1725 μg/mL suggest good to moderate activity of plant extract. Please correct.
Author Response
Reviewer 1
The authors examined the antibacterial activity of fruit extracts of P. emblica against Staphylococcus aureus, Escherichia coli, and Klebsiella pneumoniae, along with the antibiotic-resistant variants methicillin-resistant S. aureus (MRSA), extended-spectrum β-lactamase (ESBL) E. coli, and ESBL K. pneumoniae. Study results demonstrated that fruit extracts of P. emblica inhibited the growth of resistant and susceptible bacterial strains and enhanced the efficacy of conventional antibiotics. This study suggests that P. emblica extracts should be considered candidates for antibiotic development, highlighting the need for further mechanistic and phytochemical investigations. The study is interesting, scientifically important, and deserves attention.
We thank the reviewer for the positive comments. We have addressed the comments individually below.
Comments:
- Please correct the typo error in line 156- „strains. “
This has been corrected, as per the reviewer’s comment.
- It is unclear why the authors use the ALA test for toxicity assessment. ALA test is suitable for assessing toxicity from an environmental perspective, but direct applicability to human health risk assessment is limited. The authors should consider excluding that analysis from the manuscript.
The ALA test is relatively widely used as a model for toxicity evaluation. It is used as a preliminary model (as are cell line assays) and has been shown to generally produce similar results to mammalian cell cultures [1-5]. Those studies have reported that Artemia nauplii are suitable for acute toxicity testing (as it was used in our study). It is considered to be applicable to the evaluation of toxicity in humans (not just in environmental samples). Indeed, it has been shown that the ALA may be more sensitive against some toxins, compared to mammalian cells (Vero cells were used in that study) [2]. However, The ALA is a simpler and cheaper assay that can be readily done in most laboratories. As such, we believe it provides a reasonable predictor of toxicity in humans without requiring mammalian cell cultures and that it should remain in the manuscript.
- Libralato G, Prato E, Migliore L, Cicero AM, Manfra L. A review of toxicity testing protocols and endpoints with Artemia spp. Ecological indicators. 2016 Oct 1;69:35-49.
- Charoeythornkhajhornchai P, Kunjiek T, Chaipayang S, Phosri S. Toxicity assessment of bioplastics on brine shrimp (Artemia franciscana) and cell lines. Emerging Contaminants. 2023 Dec 1;9(4):100253.
- Aguirre-García YL, Castillo-Manzanares A, Palomo-Ligas L, Ascacio-Valdés JA, Campos-Múzquiz LG, Esparza-González SC, Rodríguez-Herrera R, Nery-Flores SD. Toxicity evaluation of a polyphenolic extract from flourensia cernua DC through artemia lethality assay, hemolytic activity, and acute oral test. Journal of Toxicology. 2024;2024(1):2970470.
- Zhang Y, Song S, Zhang B, Zhang Y, Tian M, Wu Z, Chen H, Ding G, Liu R, Mu J. Comparison of short-term toxicity of 14 common phycotoxins (alone and in combination) to the survival of brine shrimp Artemia salina. Acta Oceanologica Sinica. 2023 Feb;42(2):134-41.
- Chan W, Shaughnessy AE, van den Berg CP, Garson MJ, Cheney KL. The validity of brine shrimp (Artemia sp.) toxicity assays to assess the ecological function of marine natural products. Journal of Chemical Ecology. 2021 Nov;47(10):834-46.
We understand the reviewer’s concern and we acknowledge that mammalian cell line assays, or in vivo studies are often used instead for this purpose. Due to the global trend to minimise in vivo studies (and the restrictions of our institution), in vivo models were not considered. Whilst mammalian cell line assays are also used (including in our group), these have similar criticisms to the ALA assay i.e. that they are a model system and do not necessarily reflex the effects in a whole animal system. The ALA has the advantage that the extracts are being tested for general toxicity in an entire living model, rather than just a single cell line. Additionally, unless an applicable cell line(s) is selected, the results have low relevance. As our study examined toxicity in general, using cell line assays would necessitate testing against many cells from different tissues. Furthermore, the commercially available cell lines are immortalised (e.g. fibroblasts), which is also a departure from normal cells, and therefore does not accurately represent toxicity in ‘normal cells’. For this reason, we generally use ALA for crude extract studies and hold cell line assays for pure isolated compounds.
-Lines 314-316. According to the authors' criteria, MIC values from 158 μg/mL to 1725 μg/mL suggest good to moderate activity of plant extract. Please correct.
This has been corrected in the manuscript, as follows:
“…all extracts from P. emblica exhibited good to moderate antibacterial activity against S. aureus and MRSA, with MIC ranging from 158 µg/mL to 1725 µg/mL”
Reviewer 2 Report
Comments and Suggestions for Authors
The study compares the antimicrobial properties of known commercial antibiotics (often quite specialized) and aqueous, methanol, and ethanol extracts from the fruits of the plant Phyllanthus emblica. The manuscript is written in the style of a laboratory report. It has a huge introduction and an equally huge discussion, often not related to the specific results obtained. In general, the motivation for the manuscript is unclear. The concentration of the antibiotics participating in the comparison is 0.01 mg / ml, for the extracts 40.5, 47.3, and 6.9 mg / ml (water, methanol, ethanol). That is, the antibiotics were added in a concentration 3 orders of magnitude less than the extracts. At the same time, the antibiotics were often several times more effective. If we talk about MIC, the MIC of antibiotics is 3-4 orders of magnitude less than the MIC of extracts. In general, under certain conditions, antibiotics can be 6-7 orders of magnitude more effective than extracts (1,000,000 - 10,000,000 times). It seems to me that talk about a new type of antibiotics is greatly premature. The toxicity of the extracts was assessed by the survival rate of small lake crustaceans. The manuscript does not contain any specific results obtained in the experiment... I do not understand why this is necessary. Usually, when toxicity or non-toxicity of a chemical compound or mixture of chemical compounds is proven, at least tests are used on human and mammalian cell lines. With this formulation of the problem, extracts can be recommended for the treatment of acute bacterial infections only in freshwater small crustaceans... The manuscript also contains a chromatographic separation of the extracts, but no specific actions were taken with the separation products. Thus, 3 extracts were tested for MIC and ZOI in several bacterial species; toxicity was tested on crustaceans; the composition was determined using chromatography (but for some reason in %). I believe that it is premature to publish, since there is no clear motivation and the results obtained are clearly insufficient for publication.
Author Response
Reviewer 2
The study compares the antimicrobial properties of known commercial antibiotics (often quite specialized) and aqueous, methanol, and ethanol extracts from the fruits of the plant Phyllanthus emblica.
The manuscript is written in the style of a laboratory report.
We are uncertain as to what is meant by this comment. If the reviewer is stating that the manuscript is written as an original article style manuscript, then we agree, and this is the class of manuscript that we submitted it as. The manuscript is written as a research article as per numerous previous publications in this area, from multiple research groups. Additionally, all of the other reviewers were happy with the style of the manuscript.
It has a huge introduction and an equally huge discussion, often not related to the specific results obtained.
We agree with the reviewer. As such we have removed large sections of the Introduction. This has substantially reduced the introduction (from 8 paragraphs to 4). However, we believe that the information in the discussion is required to fully explain the significance and relevance of this study.
In general, the motivation for the manuscript is unclear.
The manuscript aimed to identify potential targets for further antibiotic development. In particular, we examined extracts for the ability to inhibit antibiotic resistant strains. As the development of antibiotic resistance is becoming increasingly common (and current antibiotics are therefore losing efficacy), new antibiotics are urgently required. The previous methods of antibiotic discovery (from microorganisms, semi synthesis from existing compounds) are failing to provide a viable antibiotic pipeline. Therefore, considerable research now focusses on discovery of new compounds with novel activities from other sources. An increasing number of studies are examining traditional plant-based medicines as antibiotic development targets. This has already been explained in the introduction section.
The concentration of the antibiotics participating in the comparison is 0.01 mg / ml, for the extracts 40.5, 47.3, and 6.9 mg / ml (water, methanol, ethanol). That is, the antibiotics were added in a concentration 3 orders of magnitude less than the extracts.
The assays that the reviewer mentions here are the susceptibility assays (agar diffusion assays). We believe that the reviewer has a misunderstanding of these assays and their purpose. These assays are susceptibility assays only and indicate that the extracts are worthy of the later quantification studies. They are not (and were not meant to be) quantitative assays. Therefore, comparing the concentrations tested should not be used as a correlation of potency. It is the MIC values that are determined in the liquid dilution assays that actually provide a relative comparison of potency. Indeed, the first paragraph of the discussion section already explains why this is the case.
At the same time, the antibiotics were often several times more effective. If we talk about MIC, the MIC of antibiotics is 3-4 orders of magnitude less than the MIC of extracts. In general, under certain conditions, antibiotics can be 6-7 orders of magnitude more effective than extracts (1,000,000 - 10,000,000 times).
Again, we believe that the reviewer is referring to the agar diffusion results (discussed above), as they are the results for which the amounts tested were of the magnitude differences that the reviewer states. In evaluating the relative potency of the extracts, it is the MIC values that are important, not the agar diffusion results.
Additionally, we believe that the reviewer may have missed a key issue that is very relevant to this point: The antibiotics tested in this study were tested in their pure form, whereas the extracts contain a complex mixture of phytochemicals. Indeed, it is likely due to the complexity of the extracts that the bioactive component(s) are present is substantially <1% of the total compounds in the extracts. This accounts for the differences in apparent potency that the reviewer has highlighted. For these reasons, MICs of pure antibiotics in this assay that are >1 µg/mL are generally taken to indicate (partial) resistance towards that antibiotic, whereas, the cutoffs for the extracts are very different: inactive (MIC > 10,000 µg/mL), low activity (2000–5000 µg/mL), moderate activity (1000–2000 µg/mL), noteworthy activity (400–1000 µg/mL), and good activity (100–400 µg/mL). These are accepted cutoff levels. We have highlighted this in the manuscript, and this has been reinforced in the footnote to Table 1.
It seems to me that talk about a new type of antibiotics is greatly premature.
The reviewer has misunderstood the purpose of this study, as well as its relevance. The study aimed to verify the antibiotic properties of a plant that has been traditionally used to treat bacterial infections. We examined antibiotic-resistant antibiotic strains for greater relevance. We evaluated the direct ability of the extracts to inhibit bacterial growth and found some noteworthy results. However, we tested crude extracts to determine whether they are viable targets for antibiotic discovery. Nowhere did we state that the extracts are a new type of antibiotic. Instead, we stated that they have antibiotic properties against antibiotic-resistant strains, and that they may therefore be useful targets for antibiotic discovery studies.
Furthermore, given that the extracts worked as well in β-lactam resistant bacterial strains as in the antibiotic-sensitive strains, it was believed that the extracts may contain compounds that have potentiation activity, which they did. Indeed, the extracts restored the activity of the antibiotics, even in bacteria otherwise resistant to their effects. This further reinforces the potential of these extracts for the development of novel antibiotic therapies.
The toxicity of the extracts was assessed by the survival rate of small lake crustaceans. The manuscript does not contain any specific results obtained in the experiment... I do not understand why this is necessary. Usually, when toxicity or non-toxicity of a chemical compound or mixture of chemical compounds is proven, at least tests are used on human and mammalian cell lines. With this formulation of the problem, extracts can be recommended for the treatment of acute bacterial infections only in freshwater small crustaceans...
To address the reviewer’s first point: no toxicity results were provided as all extracts had LC50 values > of 1000 µg/mL. By definition (the reference is provided in the statement below), the extracts are therefore deemed to be nontoxic. This has already been stated in section 3.4 (and shown below):
“The toxicity of plant extracts was evaluated using Artemia franciscana lethality assays. The extracts were classified as toxic if they yielded LC50 values of 1000 µg/mL or lower following a 24-hour exposure period [19]. The results for all extracts were comparable to those of the artificial seawater negative control (p > 0.05), as they did not cause ≥50% mortality at 1000 µg/mL and were therefore classified as non-toxic.”
Regarding the use of the ALA assay:
The ALA test is relatively widely used as a model for toxicity evaluation. It is used as a preliminary model (as are cell line assays) and has been shown to generally produce similar results to mammalian cell cultures [1-5]. Those studies have reported that Artemia nauplii are suitable for acute toxicity testing (as it was used in our study). It is considered to be applicable to the evaluation of toxicity in humans (not just in environmental samples). Indeed, it has been shown that the ALA may be more sensitive against some toxins, compared to mammalian cells (vero cells were used in that study) [2]. However, The ALA is a simpler and cheaper assay that can be readily done in most laboratories. As such, we believe it provides a reasonable predictor of toxicity in humans without requiring mammalian cell cultures and that it should remain in the manuscript.
- Libralato G, Prato E, Migliore L, Cicero AM, Manfra L. A review of toxicity testing protocols and endpoints with Artemia spp. Ecological indicators. 2016 Oct 1;69:35-49.
- Charoeythornkhajhornchai P, Kunjiek T, Chaipayang S, Phosri S. Toxicity assessment of bioplastics on brine shrimp (Artemia franciscana) and cell lines. Emerging Contaminants. 2023 Dec 1;9(4):100253.
- Aguirre-García YL, Castillo-Manzanares A, Palomo-Ligas L, Ascacio-Valdés JA, Campos-Múzquiz LG, Esparza-González SC, Rodríguez-Herrera R, Nery-Flores SD. Toxicity evaluation of a polyphenolic extract from flourensia cernua DC through artemia lethality assay, hemolytic activity, and acute oral test. Journal of Toxicology. 2024;2024(1):2970470.
- Zhang Y, Song S, Zhang B, Zhang Y, Tian M, Wu Z, Chen H, Ding G, Liu R, Mu J. Comparison of short-term toxicity of 14 common phycotoxins (alone and in combination) to the survival of brine shrimp Artemia salina. Acta Oceanologica Sinica. 2023 Feb;42(2):134-41.
- Chan W, Shaughnessy AE, van den Berg CP, Garson MJ, Cheney KL. The validity of brine shrimp (Artemia sp.) toxicity assays to assess the ecological function of marine natural products. Journal of Chemical Ecology. 2021 Nov;47(10):834-46.
We understand the reviewer’s concern and we acknowledge that mammalian cell line assays, or in vivo studies are often used instead for this purpose. Due to the global trend to minimise in vivo studies (and the restrictions of our institution), in vivo models were not considered. Whilst mammalian cell line assays are also used (including in our group), these have similar criticisms to the ALA assay i.e. that they are a model and system and do not necessarily reflex the effects in a whole animal system. The ALA has the advantage that the extracts are being tested for general toxicity in an entire living model, rather than just a single cell line. Additionally, unless an applicable cell line(s) is selected, the results have low relevance. As our study examined toxicity in general, using cell line assays would necessitate testing against many cells from different tissues. Furthermore, the commercially available cell lines are immortalised (e.g. fibroblasts), which is also a departure from normal cells, and therefore does not accurately represent toxicity in ‘normal cells’. For this reason, we generally use ALA for crude extract studies, and hold cell line assays for pure isolated compounds.
The manuscript also contains a chromatographic separation of the extracts, but no specific actions were taken with the separation products. Thus, 3 extracts were tested for MIC and ZOI in several bacterial species; toxicity was tested on crustaceans; the composition was determined using chromatography (but for some reason in %).
The reviewer has misunderstood the methodology that we used. We did not utilise a bioactivity driven separation methodology, nor did we isolate all of the individual components, of which there were hundreds (see Supplementary data). Instead, we utilised a metabolomic LC-MS approach to identify (rather than isolate) as many components as possible. Metabolomic profiling is a powerful tool used to identify as many as possible compounds in a crude mixture, to allow the investigator to highlight noteworthy compounds for further studies.
Reviewer 3 Report
Comments and Suggestions for Authors
The article titled "Phyllanthus emblica: Phytochemistry, Antimicrobial Potential with Antibiotic Enhancement and Toxicity Insights" investigates the antibacterial activity of various extracts of Phyllanthus emblica against both antibiotic-susceptible and resistant bacterial strains. The study employs a comprehensive methodology, including disc diffusion and liquid microdilution assays, along with LC-MS analysis to characterize the phytochemical profiles. The findings suggest that P. emblica extracts not only exhibit significant antibacterial activity but also enhance the efficacy of conventional antibiotics, addressing the critical issue of antibiotic resistance.
Specific Comments on the Article:
- Reference Strains: The specific ATCC type of the bacterial reference strains should be clearly stated.
- Limitations of Toxicity Evaluation: While the Artemia franciscana nauplii model allows for a rapid assessment of acute toxicity, it does not fully reflect the toxicity characteristics in mammalian cells. It is recommended to include additional in vitro toxicity assays using mammalian cell lines (e.g., HepG2, HEK293) to provide a more comprehensive safety evaluation.
- Insufficient Mechanistic Study: The article hypothesizes that the extract enhances antibiotic efficacy by inhibiting efflux pumps or β-lactamase activity; however, direct evidence supporting this claim is lacking (e.g., efflux pump activity assays or gene expression analysis). To strengthen the hypothesis, further validation experiments should be included, such as fluorescence-based substrate accumulation assays and qPCR analysis of resistance gene expression.
- Complexity of Extracted Compounds: The extract consists of a mixture of multiple compounds, making it difficult to determine which specific component(s) contribute to the observed antibacterial activity. Further fractionation and bioassay-guided isolation are recommended to identify the active constituents.
Author Response
Reviewer 3
The article titled "Phyllanthus emblica: Phytochemistry, Antimicrobial Potential with Antibiotic Enhancement and Toxicity Insights" investigates the antibacterial activity of various extracts of Phyllanthus emblica against both antibiotic-susceptible and resistant bacterial strains. The study employs a comprehensive methodology, including disc diffusion and liquid microdilution assays, along with LC-MS analysis to characterize the phytochemical profiles. The findings suggest that P. emblica extracts not only exhibit significant antibacterial activity but also enhance the efficacy of conventional antibiotics, addressing the critical issue of antibiotic resistance.
Specific Comments on the Article:
- Reference Strains: The specific ATCC type of the bacterial reference strains should be clearly stated.
The ATCC numbers have now been included for each of the species required.
- Limitations of Toxicity Evaluation: While the Artemia franciscana nauplii model allows for a rapid assessment of acute toxicity, it does not fully reflect the toxicity characteristics in mammalian cells. It is recommended to include additional in vitro toxicity assays using mammalian cell lines (e.g., HepG2, HEK293) to provide a more comprehensive safety evaluation.
The ALA test is relatively widely used as a model for toxicity evaluation. It is used as a preliminary model (as are cell line assays) and has been shown to generally produce similar results to mammalian cell cultures [1-5]. Those studies have reported that Artemia nauplii are suitable for acute toxicity testing (as it was used in our study). It is considered to be applicable to the evaluation of toxicity in humans (not just in environmental samples). Indeed, it has been shown that the ALA may be more sensitive against some toxins, compared to mammalian cells (vero cells were used in that study) [2]. However, The ALA is a simpler and cheaper assay that can be readily done in most laboratories. As such, we believe it provides a reasonable predictor of toxicity in humans without requiring mammalian cell cultures and that it should remain in the manuscript.
- Libralato G, Prato E, Migliore L, Cicero AM, Manfra L. A review of toxicity testing protocols and endpoints with Artemia spp. Ecological indicators. 2016 Oct 1;69:35-49.
- Charoeythornkhajhornchai P, Kunjiek T, Chaipayang S, Phosri S. Toxicity assessment of bioplastics on brine shrimp (Artemia franciscana) and cell lines. Emerging Contaminants. 2023 Dec 1;9(4):100253.
- Aguirre-García YL, Castillo-Manzanares A, Palomo-Ligas L, Ascacio-Valdés JA, Campos-Múzquiz LG, Esparza-González SC, Rodríguez-Herrera R, Nery-Flores SD. Toxicity evaluation of a polyphenolic extract from flourensia cernua DC through artemia lethality assay, hemolytic activity, and acute oral test. Journal of Toxicology. 2024;2024(1):2970470.
- Zhang Y, Song S, Zhang B, Zhang Y, Tian M, Wu Z, Chen H, Ding G, Liu R, Mu J. Comparison of short-term toxicity of 14 common phycotoxins (alone and in combination) to the survival of brine shrimp Artemia salina. Acta Oceanologica Sinica. 2023 Feb;42(2):134-41.
- Chan W, Shaughnessy AE, van den Berg CP, Garson MJ, Cheney KL. The validity of brine shrimp (Artemia sp.) toxicity assays to assess the ecological function of marine natural products. Journal of Chemical Ecology. 2021 Nov;47(10):834-46.
We understand the reviewer’s concern and we acknowledge that mammalian cell line assays, or in vivo studies are often used instead for this purpose. Due to the global trend to minimise in vivo studies (and the restrictions of our institution), in vivo models were not considered. Whilst mammalian cell line assays are also used (including in our group), these have similar criticisms to the ALA assay i.e. that they are a model and system and do not necessarily reflex the effects in a whole animal system. The ALA has the advantage that the extracts are being tested for general toxicity in an entire living model, rather than just a single cell line. Additionally, unless an applicable cell line(s) is selected, the results have low relevance. As our study examined toxicity in general, using cell line assays would necessitate testing against many cells from different tissues. Furthermore, the commercially available cell lines are immortalised (e.g. fibroblasts), which is also a departure from normal cells, and therefore does not accurately represent toxicity in ‘normal cells’. For this reason, we generally use ALA for crude extract studies, and hold cell line assays for pure isolated compounds.
We agree with the reviewer that future studies should also evaluate the extracts against a panel of cell lines and we have included the following statement in the final paragraph of the discussion:
“Toxicity assessments using Artemia nauplii confirmed that all fruit extracts of P. emblica are non-toxic, suggesting their potential safety as antimicrobial agents. However, further evaluation using diverse mammalian cell lines is necessary to confirm their suitability for medical applications.”
- Insufficient Mechanistic Study: The article hypothesizes that the extract enhances antibiotic efficacy by inhibiting efflux pumps or β-lactamase activity; however, direct evidence supporting this claim is lacking (e.g., efflux pump activity assays or gene expression analysis). To strengthen the hypothesis, further validation experiments should be included, such as fluorescence-based substrate accumulation assays and qPCR analysis of resistance gene expression.
We did not state that our extracts have those activities, but instead simply postulated these possibilities based on the evidence-based activities of the extracts’ phytochemical constituents which may be involved. Indeed, the ability of the extracts to repurpose β-lactam antibiotics confirms that the extracts overcome the cells resistance mechanisms. We have highlighted β-lactamase and/or efflux pump inhibition as likely mechanisms as these are by far the most common bacterial resistance mechanisms to this class of antibiotics. We have discussed the need for mechanistic studies, particularly in the discussion. Fluorescence-based accumulation assays and qPCR analysis are planned experiments in future studies since they are beyond the scope of the present work, which already contained substantial amounts of data from the antibacterial assays and mass spectrometry data.
To further clarify this, we have added the following text to the end of the conclusions section:
“In particular, studies evaluating the extracts for the ability to inhibit β-lactamase enzymes and efflux mechanisms are required. Additionally, qPCR studies would be useful to determine if the expression of the enzymes and efflux proteins are affected by the extract components.”
Complexity of Extracted Compounds: The extract consists of a mixture of multiple compounds, making it difficult to determine which specific component(s) contribute to the observed antibacterial activity. Further fractionation and bioassay-guided isolation are recommended to identify the active constituents.
We agree with the reviewer. Our study has identified and highlighted multiple compounds with antibacterial potential. This data is useful to guide future experimentation, as suggested by the reviewer again here. This has already been discussed throughout the discussion section.
Reviewer 4 Report
Comments and Suggestions for Authors
microorganisms-3480523-peer-review-v1
This is an interesting paper, reporting on evaluation of plant extract as alternative/additional for the treatment of some relevant microbial pathogens. Authors have constructed interesting research project; however, some important parts needs to be adjusted, corrected, updated. The main point is missing controls, and this can be questioning the author’s observations and conclusions. Moreover, in the manuscript there is missing strain identifications for applied microbial cultures, and this is another negative point. Even more, if you will clime that you have something effective against MRSA, then you will need to show that your antimicrobial works well versus several strains of MRSA, and not only versus one strain.
29% similarity in fact is quite high. Authors will need to work on this and avoid not needed repetitions.
The introduction is very interesting and informative; however, it is very long. In my opinion, big parts of the introduction can be moved to the discussion section. Please briefly state the problem, describe important pathogens, use of the plant extracts as alternatives and state of the study. All the rest is interesting information, however, needs to be move to the discussion sections, where you can enforce your results with the needs and observations/results from similar research projects.
Regarding 2.1. Do you have any information about what approach for the production of applied plat powder was used by the producer? Was it only fruit? Or other parts as well? Please, can you provide this information in the section?
Maybe 2.2. can be presented in the principal body of the manuscript and not as supplementary section.
Lbn151: Obtained from what supplier? Please, provide this information.
Ln154: MRSA do not need to be in italics.
Ln155: Gold Coast University Hospital, please, provide city and country.
Section 2.4. Need to be providing more details and for all material and equipment used in this study to provide information for suppliers, including name of the company and address (city, state (in case of federal countries) in abbreviated form, and name of the country). In following occasions, please, provide only name of the company.
Maybe authors can look for more experienced senior colleagues to help them in the adjustments and corrections, proof their manuscript.
Sections 2.5, 2.6, 2.7 and 2.8.: more details needs to be provided.
Ln210: Will be beneficial if authors will provide strain identifications for the mentioned test microorganism.
MRSA cannot be cured with traditional penicillin-related drugs, how you can explain that you have activity form penicillin? (Figure 1)? Now my question is whether the applied MRSA is really confirmed if MRSA is.
Moreover, it will be important to have controls where only solvents used in the extraction processes will be tested for antimicrobial activity. I agree that in drying processes, you are evaporating them, however, appropriate controls where solvents were treated in same way as extractions samples.
Some of the results looks very promising, however, missing appropriate controls is raising questions regarding if the observed beneficial effects are in fact what authors want, or are just combination of the role of the plant extracts and other additional factors.
Material and Methods needs to be presented in a better way, with sufficient details and will be more appropriate if and be included in the main text of the manuscript. Please, be sure that this sections can be provided with sufficient details and appropriate controls will be included.
Results and discussion are presented with logic structure, however, some of the results are a bit controversial (as MRSA been sensitive to penicillin), and missing of positive/negative controls in several parts is questioning the interpretations of the results.
Discussion is well structured, however, authors will need to try do not repeat results in the discussion section; even maybe can be considered to combine Results and Discussion as one section, if the journal (Editor) will permit.
Concussion can be more objective and state the principal observations in current study, with maybe a focus on potential further topics to be explored related to the plant extract studied.
Author Response
Reviewer 4
This is an interesting paper, reporting on evaluation of plant extract as alternative/additional for the treatment of some relevant microbial pathogens. Authors have constructed interesting research project; however, some important parts needs to be adjusted, corrected, updated. The main point is missing controls, and this can be questioning the author’s observations and conclusions. Moreover, in the manuscript there is missing strain identifications for applied microbial cultures, and this is another negative point. Even more, if you will clime that you have something effective against MRSA, then you will need to show that your antimicrobial works well versus several strains of MRSA, and not only versus one strain.
-29% similarity in fact is quite high. Authors will need to work on this and avoid not needed repetitions.
The revisions incorporated into the manuscript in response to comments by all reviewers have reduced the matching substantially.
- The introduction is very interesting and informative; however, it is very long. In my opinion, big parts of the introduction can be moved to the discussion section. Please briefly state the problem, describe important pathogens, use of the plant extracts as alternatives and state of the study. All the rest is interesting information, however, needs to be move to the discussion sections, where you can enforce your results with the needs and observations/results from similar research projects.
We agree with the reviewer. As such we have removed large sections of the Introduction. This has substantially reduced the introduction (from 8 paragraphs to 4).
- Regarding 2.1. Do you have any information about what approach for the production of applied plat powder was used by the producer? Was it only fruit? Or other parts as well? Please, can you provide this information in the section?
We state that these are fruit extracts in the Introduction when the plant is first mentioned, and the word “fruit” is used to describe the extracts in other areas of the manuscript. However, we have now ensured it is used more consistently throughout and added the word “fruit” where appropriate. It is also stated at the beginning of Section 2.1 that the powder derives from the fruit.
We have also added further details on the production of the powder, and adjusted this in Section 2.1 as follows:
“The authenticity and purity of the plant material were verified by the provider, with the product sourced from India and the tree-ripened whole berries dried at room temperature and finely ground into the powdered form.”
- Maybe 2.2. can be presented in the principal body of the manuscript and not as supplementary section.
We have now provided a brief summary of this experimental protocol within the principal body and have removed this section from the Supplementary Materials.
- Lbn151: Obtained from what supplier? Please, provide this information.
This information has now been added.
- Ln154: MRSA do not need to be in italics.
We thank the reviewer for pointing out this error. This has been corrected.
Ln155: Gold Coast University Hospital, please, provide city and country.
This information has now been added.
Section 2.4. Need to be providing more details and for all material and equipment used in this study to provide information for suppliers, including name of the company and address (city, state (in case of federal countries) in abbreviated form, and name of the country). In following occasions, please, provide only name of the company.
We have added this information for the antibiotics that were included in the study, which had already been stated in Section 2.3. We also added the supplier information for the ATCC in Section 2.3, and added source details for other solvents, equipment and reagents in Section 2.
Maybe authors can look for more experienced senior colleagues to help them in the adjustments and corrections, proof their manuscript.
The second and third authors of the manuscript are native English speakers and have collectively published >300 peer-reviewed manuscripts across several disciplines, and as such we do not believe that any further proofing of the manuscript is necessary. We have also reviewed the paper several times further to ensure there are no grammatical errors throughout. Additionally, we have used the word processors spell and grammar checking function and have had colleagues review the language used. We are unable to detect the errors that the reviewer has indicated (apart from the minor examples already indicated by the reviewer).
Sections 2.5, 2.6, 2.7 and 2.8.: more details needs to be provided.
Further detail for Section 2.5 is included in the Supplementary Materials and thus can be accessed by the reader. The amount of text was minimised in the other sections to reduce the length of the manuscript and prevent text-matching, but detailed information on the experimentation can be found in the references that have been cited within these sections.
Ln210: Will be beneficial if authors will provide strain identifications for the mentioned test microorganism.
The ATCC strain identification numbers are included in Section 2.3. Additionally, for the single clinical isolate strain, we have provided a reference to a study that thoroughly evaluated the susceptibilities/resistances of that strain.
MRSA cannot be cured with traditional penicillin-related drugs, how you can explain that you have activity form penicillin? (Figure 1)? Now my question is whether the applied MRSA is really confirmed if MRSA is.
Antibiotic-resistance does not mean that an antibiotic is completely ineffective towards a bacterium. Instead, substantially higher doses of the antibiotic may be required, and even then, the effects may not be pronounced. In our study, we used bacterial strains for which the resistance has been confirmed. Indeed, nearly all of the screened bacteria were ATCC reference strains, with defined antibiotic resistance, meaning that they are verified reference strains for these species. In the in vitro assays outlined in our study, high concentrations of penicillin did lead to some inhibition of MRSA growth as seen in Figure 1. However, this inhibition was only seen at high doses, and such amounts would not be clinically effective. Furthermore, the level of inhibition observed for MRSA by penicillin G was reduced by >75% when compared directly to the sensitive S. aureus. Therefore, this confirms that this strain is resistant (rather than totally unaffected) to that antibiotic. This was also the case for amoxicillin and oxacillin, which are other penicillin class antibiotics, and this is stated in paragraph two of the Discussion.
Moreover, it will be important to have controls where only solvents used in the extraction processes will be tested for antimicrobial activity. I agree that in drying processes, you are evaporating them, however, appropriate controls where solvents were treated in same way as extractions samples.
We agree that the solvent itself could be processed in the same way as the solvent mixed with the extracts, but this was not performed. Instead, we used the resuspension solvent (1% DMSO) as the control for the antibacterial assays, as stated in our response to the Reviewer below.
Some of the results looks very promising, however, missing appropriate controls is raising questions regarding if the observed beneficial effects are in fact what authors want, or are just combination of the role of the plant extracts and other additional factors.
We included all of the necessary controls for each of the assays. The disc diffusion assays included 1% DMSO as the negative control for the extracts, whilst blank discs infused with sterile water (containing 1% DMSO) was included for the antibiotics. The antibiotics tested in those assays acted as positive controls. For the MIC assays, again 1% DMSO was used as the negative control for extracts, whilst sterile water (containing 1% DMSO) was used for the antibiotics. The brine shrimp assays incorporated seawater as the negative controls, whilst sodium azide acted as the positive controls. All assays therefore include negative controls, as well as at least one (and generally several) positive controls. Thus, we believe we have used the appropriate controls for all assays.
Material and Methods needs to be presented in a better way, with sufficient details and will be more appropriate if and be included in the main text of the manuscript. Please, be sure that this sections can be provided with sufficient details and appropriate controls will be included.
As stated previously, we decided to reduce the amount of text in some of these sections to reduce the length of the manuscript and prevent text-matching. We believe that to describe standardised methods again in full would be redundant and would unnecessarily lengthen the manuscript for little benefit. However, detailed information on the experimentation methodology can be found within the references that have been cited within these sections.
Results and discussion are presented with logic structure, however, some of the results are a bit controversial (as MRSA been sensitive to penicillin), and missing of positive/negative controls in several parts is questioning the interpretations of the results.
As per our previous responses to comments by this Reviewer, we have responded to the findings of the MRSA response to the penicillins as well as the positive and negative controls that were used in the assays. As such, we believe we have addressed these points.
Discussion is well structured, however, authors will need to try do not repeat results in the discussion section; even maybe can be considered to combine Results and Discussion as one section, if the journal (Editor) will permit.
We agree with the Reviewer in that there is some repetition of results in the Discussion, but this arose simply because we followed the Microorganisms template which has separate Results and Discussion sections. However, we do not believe that it is necessary to combine the two sections and any results that are repeated in the Discussion are done with brevity whilst maintaining clarity about which results we are discussing.
Concussion can be more objective and state the principal observations in current study, with maybe a focus on potential further topics to be explored related to the plant extract studied.
We have added content on the future directions of the project to the Conclusions, as follows:
“In particular, studies evaluating the extracts for the ability to inhibit β-lactamase enzymes and efflux mechanisms are required. Additionally, qPCR studies would be useful to determine if the expression of the enzymes and efflux proteins are affected by the extract components.”
Reviewer 5 Report
Comments and Suggestions for Authors
To correct/complete to improve some small parts of the manuscript:
Introduction – is too long, with many details, especially the paragraphs containing information about bacterial resistance and the part of the studies conducted on Phyllanthus, which could be moved to Discussions.
Extract preparation – to be completed briefly. Also, specify the samples you worked on and their abbreviations.
HLC-MS method – to specify the working conditions. How the identification and also the quantification of the components was done.
Specify how many samples you took into work.
In table 3 – the structural class could also be included.
Author Response
Reviewer 5
To correct/complete to improve some small parts of the manuscript:
Introduction – is too long, with many details, especially the paragraphs containing information about bacterial resistance and the part of the studies conducted on Phyllanthus, which could be moved to Discussions.
We agree with the Reviewer. As such we have removed large sections of the Introduction. This has substantially reduced the introduction (from 8 paragraphs to 4).
Extract preparation – to be completed briefly. Also, specify the samples you worked on and their abbreviations.
This has now been included in Section 2.2, and the extract sample names were specified with their abbreviations.
HLC-MS method – to specify the working conditions. How the identification and also the quantification of the components was done. Specify how many samples you took into work.
For brevity, we decided to reduce the amount of text in this section only to the most pertinent information. However, we have cited a paper that contains a more detailed description of the protocol that we employed, and the reader has been referred to that study for a complete description of the method. We believe that repeating the entire method in our manuscript is redundant and would be unnecessarily repetitious.
In table 3 – the structural class could also be included.
Whilst we believe that defining the molecular class of all compounds is not required, we do agree with the reviewer that it would be useful to highlight tannin and flavonoid compounds as these have been highlighted in the manuscript for their potential antibacterial activity. We have now highlighted these with different colours in Table 3, and we have provided the relevant explanation below the table.
Round 2
Reviewer 2 Report
Comments and Suggestions for Authors
The manuscript contains a huge discussion, often unrelated to the results obtained. This needs to be corrected! The extracts that the authors study have significantly higher concentrations compared to antibiotics and have significantly smaller effects. I think the authors should write in the abstract that the studied extracts are significantly less effective compared to commercial antibiotics. It seems to me that the authors' reasoning about toxicological models is extremely far from the truth. There are quite a large number of chemical compounds that are poisonous for crustaceans and insects, but do not have a significant effect on mammals. I think the authors are familiar with the achievements of chemical science sold in any store. The authors claim that their team is working with eukaryotic cell cultures. In light of this, it will not be difficult for the authors to conduct toxicological tests on human cell cultures.
Author Response
The manuscript contains a huge discussion, often unrelated to the results obtained. This needs to be corrected!
The reviewer has stated their belief that the discussion is too long and needs to be shortened. We believe that it is not possible to address reviewer 2’s comment and still comply with the comments of two other reviewers in the first round of review. Indeed, two other reviewers requested that the discussion section be increased by the addition of some of the details that we deleted from the introduction section. Thus, if we decreased the discussion in line with reviewer 2’s comment, it would negate the reviews of two other reviewers.
Additionally, we disagree with the reviewer that statement that large sections of the results are unrelated to the results obtained. Indeed:
- The first paragraph states the significance of the project and justifies the inclusion of the β-lactam resistant bacterial strains and justifies their inclusion. This is fundamental to the significance of the study.
- The second paragraphs relate the results obtained to their importance as potential drug development against antibiotic resistant bacteria.
- The third and fourth paragraphs explore the resistance mechanisms used by the β-lactam resistant bacterial strains, which is crucial to understanding how the extracts may elicit their effects. The activities of the extracts are correlated with these resistance mechanisms in fourth paragraph.
- The fifth paragraph explains the significance of combinational antibiotic therapies and their relevance to future antibiotic therapies, whilst the sixth and seventh paragraphs explain the results obtained in the context of combinational effects.
- The eighth and ninth paragraph relate the combinational effects to bacterial resistance mechanisms to explain the effects.
- The nine through the second last paragraph discuss the LC-MS phytochemical studies and relate the identified compounds to previous reports of antibacterial related activities of those compounds (and closely related compounds).
- The final paragraph discusses the toxicity evaluations, which are important As they indicate the safety of the extracts for therapeutic use.
Throughout all paragraphs in the discussion section, we also indicate future research that is required as a result of this study.
As such, we fail to see how the discussion is unrelated to our study, as reviewer 2 states.
The extracts that the authors study have significantly higher concentrations compared to antibiotics and have significantly smaller effects. I think the authors should write in the abstract that the studied extracts are significantly less effective compared to commercial antibiotics.
We assume that the reviewer works with pure compounds rather than extracts, because the fact that pure compounds have greater potency that crude extracts is always the case. Indeed, if a crude extract was reported to have similar potency as measured by MIC values, then that would be cause for suspicion. Crude extracts contain hundreds (or even thousands) of individual components, with only a small percentage of those (generally <1%) contributing to the antibiotic properties of the extract. Thus, we would be sceptical of any extract that was claimed to have similar MIC values to the pure compounds. For this reason, the cutoff values for potency ranges are different for extracts and pure compounds (several orders of magnitude). This was already explained in the original manuscript, and we outlined the reasons in detail in our response to the initial round of review by this reviewer. Furthermore, the reviewer has requested that we put a statement in the abstract highlighting the differences in MIC between the extracts and pure compounds. This has already been highlighted and explained throughout the manuscript. To require that a statement that the pure compounds were more potent that the crude extracts would be akin to requiring a statement that water is wet. Both facts are self-evident.
It seems to me that the authors' reasoning about toxicological models is extremely far from the truth. There are quite a large number of chemical compounds that are poisonous for crustaceans and insects, but do not have a significant effect on mammals. I think the authors are familiar with the achievements of chemical science sold in any store. The authors claim that their team is working with eukaryotic cell cultures. In light of this, it will not be difficult for the authors to conduct toxicological tests on human cell cultures.
This point has already been explained. This is a widely used assay, and it generally correlates well to the cell assays for most toxins. Furthermore, cell assays suffer from several issues: they utilise immortalised cell lines, and therefore to not relate to the situation in vivo. Additionally, as we wanted a general evaluation of toxicity (rather than toxicity in a particular cell line) as antibiotic treatments would generally be systemic, an understanding of toxicity in (for example) immortalised skin cells would not be relevant to (for example) hepatotoxicity. For a realistic understanding of toxicity, numerous cell lines (or live animals) would be required. It is not ethical to undertake animal experiments until research has advanced substantially, and we believed that a general assessment of toxicity was better in this context than an assessment against limited (and possibly not relevant) cell lines.
Reviewer 3 Report
Comments and Suggestions for Authors
Thank you for your response. Your answer has addressed the concerns satisfactorily.
Author Response
We thank the reviewer for this positive response.
Reviewer 4 Report
Comments and Suggestions for Authors
Authors have improved the manuscript and can be suggested for publication
Author Response

(The authors gave the same response as above.)
